# Safety and Efficacy of Crizotinib in Combination with Temozolomide and Radiotherapy in Patients with Newly Diagnosed Glioblastoma: Phase Ib GEINO 1402 Trial

**DOI:** 10.3390/cancers14102393

**Published:** 2022-05-12

**Authors:** María Martínez-García, Guillermo Velasco, Estela Pineda, Miguel Gil-Gil, Francesc Alameda, Jaume Capellades, Mari Cruz Martín-Soberón, Israel López-Valero, Elena Tovar Ambel, Palmira Foro, Álvaro Taus, Montserrat Arumi, Aurelio Hernández-Laín, Juan Manuel Sepúlveda-Sánchez

**Affiliations:** 1Medical Oncology Department, Hospital del Mar, 08003 Barcelona, Spain; ataus@psmar.cat; 2Medical Oncology Department, Centro Integral Oncológico Clara Campal HM Delfos, 08023 Barcelona, Spain; 3Cancer Research Program, Instituto Hospital del Mar de Investigaciones Médicas, 08003 Barcelona, Spain; 4Biochemistry and Molecular Biology Department, School of Biology, Complutense University, 28040 Madrid, Spain; gvelasco@ucm.es (G.V.); ilvalero@ucm.es (I.L.-V.); elenatov@ucm.es (E.T.A.); 5Instituto de Investigación Sanitaria San Carlos (IdISSC), 28040 Madrid, Spain; 6Translational Genomics and Targeted Therapeutics in Solid Tumors, August Pi i Sunyer Biomedical Research Institute (IDIBAPS), Medical Oncology Department, Hospital Clínic, 08036 Barcelona, Spain; epineda@clinic.cat; 7Medical Oncology Department, Institut Català d’Oncologia L’Hospitalet, 08908 L’Hospitalet de Llobregat, Spain; mgilgil@iconcologia.net (M.G.-G.); marumi@psmar.cat (M.A.); 8Pathology Department, Hospital del Mar, Universitat Autònoma de Barcelona, 08193 Barcelona, Spain; 86780@parcdesalutmar.cat; 9Radiology Department, Hospital del Mar, 08003 Barcelona, Spain; jcapellades@parcdesalutmar.cat; 10Medical Oncology Department, Hospital Universitario 12 de Octubre, 28041 Madrid, Spain; mcms.207@gmail.com; 11Radiation Oncology Department, Hospital del Mar, 08003 Barcelona, Spain; pforo@parcdesalutmar.cat; 12Pathology Department (Neuropathology), Hospital 12 de Octubre Research Institute (imas12), 28041 Madrid, Spain; aurelio.hlain@salud.madrid.org

**Keywords:** glioblastoma, crizotinib, temozolomide, radiotherapy, midkine

## Abstract

**Simple Summary:**

Most patients with glioblastoma, the most frequent primary brain tumor in adults, develop resistance to standard first-line treatment combining temozolomide and radiotherapy. Signaling through the hepatocyte growth factor receptor (c-MET) and the midkine (ALK ligand) promotes gliomagenesis and glioma stem cell maintenance, contributing to the resistance of glioma cells to anticancer therapies. This trial reports for the first time that the addition of crizotinib, an ALK, ROS1, and c-MET inhibitor, to standard RT and TMZ is safe and resulted in a promising efficacy for newly diagnosed patients with glioblastoma.

**Abstract:**

Background: MET-signaling and midkine (ALK ligand) promote glioma cell maintenance and resistance against anticancer therapies. ALK and c-MET inhibition with crizotinib have a preclinical therapeutic rationale to be tested in newly diagnosed GBM. Methods: Eligible patients received crizotinib with standard radiotherapy (RT)/temozolomide (TMZ) followed by maintenance with crizotinib. The primary objective was to determine the recommended phase 2 dose (RP2D) in a 3 + 3 dose escalation (DE) strategy and safety evaluation in the expansion cohort (EC). Secondary objectives included progression-free (PFS) and overall survival (OS) and exploratory biomarker analysis. Results: The study enrolled 38 patients. The median age was 52 years (33–76), 44% were male, 44% were MGMT methylated, and three patients had IDH1/2 mutation. In DE, DLTs were reported in 1/6 in the second cohort (250 mg/QD), declaring 250 mg/QD of crizotinib as the RP2D for the EC. In the EC, 9/25 patients (32%) presented grade ≥3 adverse events. The median follow up was 18.7 months (m) and the median PFS was 10.7 m (95% CI, 7.7–13.8), with a 6 m PFS and 12 m PFS of 71.5% and 38.8%, respectively. At the time of this analysis, 1 died without progression and 24 had progressed. The median OS was 22.6 m (95% CI, 14.1–31.1) with a 24 m OS of 44.5%. Molecular biomarkers showed no correlation with efficacy. Conclusions: The addition of crizotinib to standard RT and TMZ for newly diagnosed GBM was safe and the efficacy was encouraging, warranting prospective validation in an adequately powered, randomized controlled study.

## 1. Introduction

Glioblastoma (GBM) is the most frequent primary brain tumor in adults. Despite newly biological discoveries, the prognosis of GBM remains dismal. The median survival ranges from 14 to 21 months when treated with standard radiation and chemotherapy after surgical excision [1,2]. Molecular prognostic factors include: the O6-methylguanine DNA methyltransferase (MGMT) promoter methylation status [3], and mutations in the isocitrate dehydrogenase 1 and 2 (IDH) genes [4,5]. Efforts are underway to identify molecular pathways involved in GBM resistance to standard chemoradiation [6,7,8,9,10].

Hepatocyte growth factor receptor (c-MET) signaling has a role in gliomagenesis and glioma stem cell (GSC) maintenance [11]. Increased signaling through the c-MET receptor, either through genomic amplification, missense mutations, inappropriate activation, or increased levels of its ligand, the hepatocyte growth factor (HGF), promotes GSC survival [12,13]. c-MET activation also triggers cellular mechanisms mediated by transcription factors such as Nanog, Sox2, c-Myc, and Oct-4, leading to cell reprogramming, dedifferentiation, and the rise of more GSCs [14].

Midkine (MDK), an anaplastic lymphoma kinase (ALK) ligand, also promotes the resistance of glioma cells to anticancer therapies such as radiotherapy (RT) and temozolomide (TMZ). Increased MDK levels are correlated with a worse prognosis in GBM patients [15,16]. MDK tumorigenic activity is mostly mediated by the stimulation of ALK receptors [15]. The ALK receptor is also expressed at a significantly high level in GBM, and chromosomal rearrangements, amplifications, and mutations of the ALK gene are commonly found and associated with poor survival and a higher tumor grade of GBM [17]. Furthermore, aberrant MDK-ALK signaling promotes the constitutive activation of AKT/MTOR cascade and contributes to the self-renewal and stemness of GSCs [18].

Pharmacological targeting of the MDK/ALK axis with crizotinib effectively acts on the population of glioma-initiating cells (GIC) in vitro and in tumor xenografts. Moreover, crizotinib enhances the response of GIC cultures to temozolomide [18] and has demonstrated intracranial efficacy [19,20]. These findings led to the present clinical trial of crizotinib in combination with TMZ and RT in newly diagnosed glioblastoma.

## 2. Material and Methods

### 2.1. Study Design

GEINO-1402 was a phase Ib, open-label, single-arm, multicenter study with an initial dose-escalation phase (NCT02270034/2014-000912-33). The safety and efficacy of crizotinib in combination with TMZ and RT in patients with newly diagnosed glioblastoma was assessed in the subsequent expansion phase. The study was sponsored by the Spanish Research Group in Neuro-Oncology (GEINO) and conducted in 4 Spanish Hospitals.

### 2.2. Study Population

Patients >18 years who were recently diagnosed with histologically confirmed glioblastoma in the 4–7 weeks prior to patient registration, with available paraffin-fixed (FFPE) or frozen tumor samples were eligible. The glioblastoma diagnosis was confirmed retrospectively by a central pathologist. Other inclusion criteria included: a Karnofsky index ≥60%, being recovered from previous surgeries (at least 4 weeks prior to starting the study treatment), and having a normal bone marrow, hepatic, and renal function. Patients were excluded if they had metastatic extracranial disease, a GLIADEL implant, clinically significant gastrointestinal abnormalities, any psychiatric or cognitive disorder that interfered with the free-willing provision of informed consent, significant or not controlled cardiovascular disease, or second neoplasms. Previous treatment with chemotherapy or radiotherapy for the brain tumor was not allowed. Enzyme-inducing antiepileptic drugs must have had a washout period of at least 7 days before inclusion in the study.

### 2.3. Treatment Plan

For all patients, the study treatment began 4–7 weeks after the initial surgical procedure. Eligible patients received daily oral crizotinib in addition to standard RT and TMZ and, afterward, they continued crizotinib daily with sequential adjuvant TMZ. Further maintenance treatment with crizotinib in monotherapy beyond 6 sequential TMZ cycles was allowed at the treating physician’s discretion (Figure 1A).

The trial had an initial dose escalation (DE) phase following a 3 + 3 design to find the safe dose of crizotinib in combination with the Stupp scheme followed by an expansion cohort (EC).

### 2.4. Objectives and Assessments

During the DE, the primary objective was to determine the maximum tolerated dose (MTD) and the recommended phase 2 dose (RP2D) of crizotinib in combination with radiotherapy and temozolomide in newly diagnosed glioblastoma. Crizotinib MTD was determined using a standard “3 + 3” dose-escalation design. Dose-limiting toxicities (DLTs) were defined as any non-hematological toxicity of grade ≥3 and any hematological toxicity of grade ≥4 observed during the first 12 weeks of therapy. The primary objective of the expansion phase was to further assess the safety of the combination at the recommended dose from the escalation phase. Secondary objectives included efficacy by means of objective response rate (ORR) according to RANO criteria, progression-free survival (PFS) and overall survival (OS), changes in the neurological status of patients using the Barthel Index and the Mini-Mental Test, and an exploratory biomarker analysis.

The primary endpoint was to establish a safety profile based on the frequency and severity of adverse events (AEs) graded according to the National Cancer Institute (NCI) Common Terminology Criteria for Adverse Events (CTCAE), Version 4.0. The disease was radiologically assessed by MRI at baseline (within the 10-day window prior to the start of treatment and at least 21 days after surgery), at 4 weeks after the completion of RT and then every 12 weeks until disease progression according to RANO criteria. MRI images were centrally reviewed.

### 2.5. Statistics

The sample size was estimated using a two-stage model, with 12 patients in the dose-escalation phase and 26 patients recruited in the expansion phase. Thus, the expected total number of patients to be recruited was 38.

Baseline, efficacy, and safety variables, as appropriate, are shown in summary tables. Descriptive statistics were used to summarize continuous variables (median, 95% confidence intervals (CI), or full-range intervals). Categorical data were summarized by frequency counts and percentages. The response percentages were estimated using 95% confidence intervals or full range intervals. The Kaplan–Meier method was used to estimate time-to-event endpoints. Cox regression analysis were used to obtain hazard ratios and CIs. Patients without documented progression or death were censored at the last date of tumor evaluation or follow-up. Statistical analyses were performed with SPSS (IBM SPSS Statistics Version 26, Armonk, NY, USA) and R (version 3.6.3 (29 February 2020) “Holding the Windsock”, The R Foundation for Statistical Computing, Vienna, Austria). RStudio (version 1.2.5033 (2009–2019), RStudio, Inc., Boston, MA, USA) and GraphPad Prism version 9 (GraphPad Software, La Jolla, CA, USA) were used to generate figures and tables. All statistical tests were two-tailed, and results were considered statistically significant if *p* < 0.05.

### 2.6. Molecular Biomarker Analysis

Molecular analysis was carried out centrally from FFPE tumor tissue samples obtained during the screening phase or prior to patient inclusion.

DNA was extracted from tumor samples with the Qiagen DNA extraction kit (Qiagen, Wetzlar, Germany) and analyzed by methylation-specific multiplexed ligation-dependent probe amplification (MS-MLPA) to evaluate the methylation status of the MGMT promoter, the amplification of EGFR, and the mutation of IDH1/2 genes as previously described.

Chromosomal alterations on ALK, c-MET, and ROS genes was assessed by FISH on FFPE slices using an ALK break-apart FISH probe kit (Abbott Molecular, Des Plaines, IL, USA), MET 7q31 SE 7 probe (Leica Biosystems, Amsterdam, The Netherlands), and ROS1 break apart FISH probe kit (Abnova, Taipei, Taiwan). All determinations were carried out following the manufacturer’s instructions.

The serum MDK levels were determined at baseline, at the time of end of radiotherapy and at the time of disease progression. The MDK level was assessed with the human MDK enzyme-linked immunosorbent assay (ELISA) Kit for human MDK detection (LYRAMID) according to the manufacturer’s briefings.

Serum MDK levels were used for correlation studies with efficacy and survival. Patients were stratified according to their MDK blood levels. Kaplan–Meier analysis was used for PFS and OS in these patient subsets and the log rank test was used for statistical comparisons between groups.

## 3. Results

Between 2014 and 2020, 38 patients from four university hospitals in Spain were enrolled: 37 were evaluable for safety (ITT cohort) and 36 for efficacy (PP cohort); 12 were enrolled in the dose-escalation phase and 26 in the expansion cohort. Patient characteristics and molecular biomarkers are summarized in Table 1.

### 3.1. Dose-Escalation Phase

For the dose-escalation phase, 12 patients were enrolled: 3 in the first cohort (crizotinib 200 mg/once a day (QD)), 6 in the second cohort (crizotinib 250 mg/QD), and 3 in the third cohort (crizotinib 200 mg/twice a day (BID), 400 mg/day) (Figure 1B).

Enrolled patients started the study treatment scheme after a median of 5.3 weeks (95% CI: 4.1–7) after surgery. Throughout the dose-escalation, all patients completed concomitant therapy, 11 patients (91.67%) initiated the sequential adjuvant treatment, and 7 of them completed the planned six cycles of adjuvant therapy and started and completed maintenance therapy with crizotinib.

All patients in the dose-escalation phase experienced treatment-related adverse events (TRAEs). The most common any-grade treatment-related adverse events consisted of: nausea (75%), asthenia (58.3%), transaminitis (50%), neutropenia (50%), alopecia (41.7%), and diarrhea (41.7%). Thirteen grade ≥3 TRAEs were reported in four (33.3%) patients: five neutropenia, two transaminitis, two thrombopenia, one lymphopenia, one constipation, one hypophosphatemia, and one asthenia (Appendix A).

There were 0/3 DLTs reported in the first cohort, 1/6 in the second cohort consisting of a grade 3 transaminitis, and 2/3 in the third cohort, which consisted of a patient that experienced a grade 3 constipation and another patient that suffered a grade 3 transaminitis followed by a grade 4 neutropenia (Table 2). Based on the safety profile, MTDs, and the observed DLTs, the regimen of crizotinib 250 mg/day combined with standard RT and TMZ was selected as the RP2D for further research in the expansion phase.

### 3.2. Dose-Expansion Phase

In the expansion phase, 26 additional patients were enrolled and 24 received the study treatment in compliance with protocol specifications. One was a screening failure and another one started treatment but discontinued because of a wrong initial diagnosis. All 24 (100%) patients that started the experimental treatment schedule completed the concomitant therapy. Sixteen (66.7%) of them completed the planned six cycles of adjuvant treatment, and thirteen (65%) of these patients completed maintenance therapy with crizotinib. Therefore, for the study as a whole regardless of the study phase, 36 (100%) patients completed concomitancy, 23 (63.9%) adjuvancy, and 20 (55.6%) maintenance (Figure 1B). Most patients discontinued treatment due to progression of the disease (24; 66.7%). Other reasons for treatment discontinuation included unacceptable toxicity (6; 16.7%), physician criteria (3; 8.3%), death (1; 2.8%), consent withdrawal (1; 2.8%), and non-treatment-related AE (1; 2.8%).

Consistently with the dose-escalation, all patients evaluable for safety (*n* = 25) in the expansion cohort experienced at least one TRAE. The most common TRAEs (all grades) included: nausea (64%), asthenia (64%), transaminase elevation (40%), anorexia (32%), vomiting (32%), thrombocytopenia (28%), neutropenia (24%), and diarrhea (24%). In the expansion phase, 9 out of 25 patients (36%) presented grade ≥3 TRAEs, the most common being transaminase elevation (20%), thrombocytopenia (8%), and lymphopenia (8%) (Table 2). Special precaution was taken with visual impairment: complete ophthalmological examination was performed at baseline, 4 weeks after the start of treatment, and at the discretion of the principal investigator thereafter. Eye disorders were infrequent and low grade: only three (8.3%) patients experienced dry eye, two (5.6%) patients experienced blurred vision, one (2.8%) patient suffered epithelitis, and one patient (2.8%) had xerophthalmia.

Considering all patients evaluable for safety (*n* = 37), most toxicities were observed during the concurrent treatment with crizotinib, RT and TMZ. In the concomitant phase, 36 (97.3%) patients presented TRAEs, and 12 (32.4%) of them suffered at least a grade ≥3 TRAE. The adjuvant phase was started by 33 patients, of which 28 patients (84.8%) presented TRAEs, and 6 (18.2%) of them experienced a grade ≥3 TRAE. The maintenance therapy was initiated by 20 patients and 7 (35%) of them presented AEs, 1 (5%) of which was a grade 3 edema (Appendix A).

### 3.3. Efficacy

At the time of this analysis, 30 (83.3%) events for PFS were reported, 28 patients had progressed, and 2 patients died without progression. Overall, 23 (63.9%) patients died throughout the study, mainly due to the inexorable progression of the disease. After a median follow-up of 18.7 months (range 2.2–61.5), the median PFS was 10.7 months (95% CI, 7.7–13.81 months), with the 6 m and 12 m PFS rates being 71.5% and 38.8%, respectively (Figure 2A). The median OS was 22.6 m (95% CI, 14.8–31.1), with 6 m and 12 m OS rates of 91.5% and 79.3%, respectively (Figure 2B). After 24 months, the OS rate was estimated at 44.5% (Figure 2B). The stratification of patients by the methylation status of the MGMT promoter showed statistically significant differences between subgroups in terms of efficacy. The median PFS was 19.1 months (95% CI: 10.2–28) in the methylated subgroup versus 7.4 months (95% CI: 4.6–10.2) in the non-methylated (*p* = 0.001), and the median OS was 31.4 months (95% CI: 15.4–47.4) versus 18.6 months (95% CI: 11.7–25.4), respectively (*p* = 0.046) (Figure 2C,D). Among those IDH1/2-wild type patients, the median OS was 18.6 m (95% CI: 11.6–25.6). The clinical evolution of each patient is depicted in Figure 2E.

The median time to best response was 2.4 months (95% CI: 2.4–2.6). Among the 36 patients analyzed for efficacy, 21 were evaluable for response, 1 (4.8%) patient had a complete response (CR), 5 (23.8%) had a partial response (PR), 13 (61.9%) had stable disease (SD), and 2 (9.5%) had progression disease (PD) as their best response to treatment (Figure 3A,B).

The corticosteroids baseline dose was consistently reduced in four (21.1%) patients and increased in five (26.3%) patients through the study. There were no statistically significant differences in the use of corticosteroids throughout the study period. Regarding the patient’s performance and cognitive status, the Barthel Index had a small but significant decrease during the safety visit, a mean of 97.7 (95% IC: 95.8–99.6) and 85.7 (95% IC: 74.5–96.9) for the baseline and safety visit, respectively (*p* = 0.023), and the Mini-Mental Test had a small but significant increase during the concomitant phase visits (week 5 and 10 after the start of treatment) when compared to the baseline, with means of 28.8 (95% CI: 27.8–29.7), 28.8 (95% CI: 28.2–29.5), and 27.9 (95% CI: 27.0–28.9), respectively (*p* = 0.047 and 0.01).

### 3.4. Molecular Biomarkers and Correlation with Efficacy

The soluble serum MDK levels were available for 26 (68.4%), 29 (76.3%), and 17 (44.7%) patients at baseline, after RT and at progression of the disease, respectively (Figure 4A). Among those, the mean MDK level at baseline was 903 ng/mL (SD: 1203), following a normal distribution. The MDK levels experienced a small but statistically significant increase after RT with a mean of 1064 ng/mL (SD: 1281) (*p* = 0.002), and this increase was maintained after disease progression, with a mean of 1270 ng/mL (SD: 1913), despite there being no statistically significant differences when compared to the baseline or after RT (*p* = 0.584 and 0.944, respectively) (Figure 4A).

Patients were stratified according to their basal serum levels of MDK in two subgroups (<1000 ng/mL and >1000 ng/mL). The cutoff value for stratification was selected arbitrarily to generate balanced groups taking into account the distribution of MDK levels in our cohort, but other cut-off values for stratification, such as the median, were tested and did not modify the current findings (data not provided). The median PFS and OS for those patients with lower MDK levels were 11.3 months (95% CI: 10.1–12.6) and 23.5 months (95% CI: 12.2–34.9); whereas for those with high MDK levels they were 6 months (95% CI: 5.4–6.5) and 15.1 months (95% CI: 12.1–18.2) (Figure 4B,C). In our cohort, there were no statistically significant differences in efficacy endpoints such as PFS and OS when patients were stratified by their basal levels of MDK in two subgroups (*p* = 0.064 and 0.329, respectively) (Figure 4B,C). There was no correlation between survival and MDK levels at baseline, the end of treatment, or progression (*p* value = 0.623, 0.678, and 0.38, respectively, using the Cox regression model) (Appendix A).

## 4. Discussion

The inhibition of cellular mechanisms involved in GSC rise and maintenance, such as the c-MET and ALK pathways, has been proposed as a therapeutic approach to overcome treatment resistance in GBM. We demonstrated for the first time that crizotinib in combination with TMZ and RT has a tolerable safety profile and promising efficacy as a first-line therapy for GBM patients.

In this study, the toxicity profile was manageable and similar to previous reports [17,18,19,20,21,22,23]. The RP2D for crizotinib in combination was set at 250 mg/day. The tolerability of crizotinib was similar to that reported for c-MET and ALK inhibitors [17,18,19,20,21,22,23]. The most frequent treatment-related adverse events were nausea, fatigue, transaminitis, and neutropenia. All events were manageable and reversible after dose reduction/interruption and or adequate treatment. Grade ≥3 events were mainly reported during the concomitant treatment phase and the most common ones were transaminitis, neutropenia, and thrombocytopenia. The toxicity profile did not differ from previous studies [17,18,19,20,21,22,23]. Interestingly, peripheral edema was less frequent when compared to previous studies with crizotinib (16.6 vs. 37.2, respectively) [17,18,19,20,21,22,23]. Only one G3 edema was reported during the maintenance phase. Elevated transaminases and a decrease in blood cell counts were common findings, especially during the concomitant phase, and transaminase increase was defined as DLT in two patients, pointing to the convenience of close monitoring for the hepatic enzymes along with the hematologic counts. Eye disorders, which are known to cause frequent toxicities causally related with crizotinib, had a low frequency and severity. No patient experienced vision loss and there were no grade ≥3 toxicities involving the eye.

Crizotinib in combination with TMZ and RT had an encouraging antitumor activity, evoking a tumor shrinkage in more than 25% of patients (i.e., 6 patients having a response among 21 evaluable patients). This data should be interpreted with caution considering this is a challenging setting for tumor response evaluation due to the potential effect of radiation-induced necrosis generating similar effects to pseudo-progressions and the evaluation of lesions after surgery leading to many non-evaluable patients. In fact, only 21 out of 36 patients were evaluable for response in our trial.

Crizotinib combined with standard TMZ and RT showed a median PFS of 10.7 months (95% CI: 7.7–13.8) and a median OS of 22.6 months (95% CI: 14.1–31.1), which are higher than the expected median PFS (6.9 months) and OS (14.6 months) from benchmark studies administering the Stupp scheme alone as a first-line treatment [1,2]. For example, the combination of RT and TMZ with iniparib, an intracellular converter of nitro radical ions, showed a median OS of 21.6 months [6], while the administration of RT and TMZ with anlotinib, a multitarget tyrosine kinase inhibitor, and vorinostat, a histone deacetylase (HDAC) inhibitor, reported a median OS of 17.4 and 16.1 months, respectively [7,8,9]. Preliminary analysis from a phase III randomized trial studying the combination of marizomib, a proteasome inhibitor, with the Stupp scheme failed to demonstrate a benefit in terms of survival, with a median OS of 15.7 months [8]. Considering our results in this context, the combination of chemoradiotherapy with crizotinib achieved a promising survival.

Despite the promising antitumor activity and survival reported in our trial, it needs to be noted that this study was small, and that patients may have presented differential baseline prognostic characteristics that made them fitter than other single-arm studies. For instance, our sample had a higher percentage of patients with MGMT methylation, a positive prognostic factor, than the phase II trial with iniparib (44.4% vs. 36%, respectively) [6]. Conversely, we observed only three (8.3%) patients with IDH mutations, which predict longer survival and response to temozolomide in GBM [24]. Given the indirect nature of the comparison, our findings will need to be validated in prospective, randomized controlled studies.

The encouraging efficacy reported in this study suggests a synergistic effect of crizotinib combined with RT and TMZ. It is postulated that the main mechanism by which the MDK/ALK axis regulates GSC biology is based on the control of the stability of SOX9 through regulation of the autophagosome-lysosome pathway [18]. Likewise the inhibition of autophagy increases susceptibility to TMZ [25]. Thus, targeting the autophagy pathway may have a relevant role in treatment resistance and might be considered for future research in newly diagnosed GBM. Moreover, next generation ALK inhibitors reach higher brain concentrations than crizotinib and may achieve even better control of the disease.

In our population, the methylation of the MGMT promoter correlated with efficacy variables, corroborating previous studies reporting that this event is a positive prognostic factor for GBM response to treatment [3,26]. The low percentage of patients with positive alterations in ALK, c-MET, and ROS genes in our population did not allow further correlation studies for efficacy.

Previous studies demonstrated that MDK levels correlate with a poor prognosis in GBM patients [15,16]. Despite a tendency in line with these reports, in our cohort there were no statistically significant associations between MDK levels at baseline and efficacy in terms of PFS and OS. Soluble MDK levels detected in serum were significantly increased after treatment, being significantly higher after the concurrent phase with RT, TMZ, and crizotinib when compared to the baseline. Thus, the MDK levels increased after disease progression, although no statistical differences were found when compared to the initial basal levels. This increase in serum MDK levels upon treatment may correspond to an adaptive mechanism that could contribute to treatment resistance. In fact, MDK has been shown to protect glioblastoma and neuroblastoma cells against cannabinoid and doxorubicin treatments, respectively [15,27]. Furthermore, MDK was overexpressed in drug-resistant gastric cancer cell sublines compared with the parental drug-sensitive ones [28]. However, the low number of patients and the great variability in MDK levels in our population makes it impossible to reach solid conclusions and further research will be needed to explore this hypothesis and corroborate the correlation between MDK and either treatment resistance or efficacy.

No clinically relevant worsening of performance, cognitive status, and/or significant increase in the corticosteroid dose were observed throughout the study.

## 5. Conclusions

In conclusion, in this phase Ib study, the inhibition of c-MET and ALK through the addition of crizotinib to standard RT and TMZ was safe and resulted in a highly promising efficacy for newly diagnosed GBM, granting further investigation on the combination of ALK/MET inhibitors with chemoradiotherapy.

## Figures and Tables

**Figure 1 cancers-14-02393-f001:**
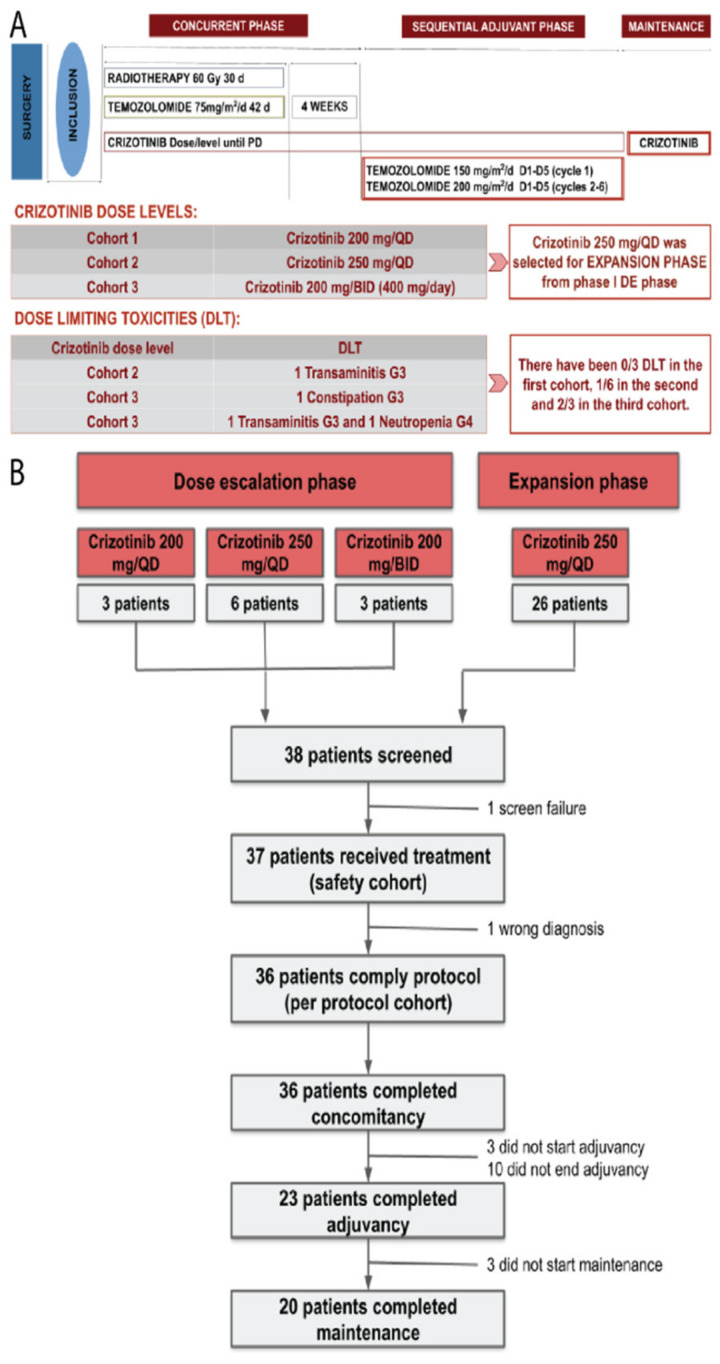
(**A**) Trial design and dose levels of the dose escalation phase of the study and (**B**) CONSORT flowchart for patient distribution.

**Figure 2 cancers-14-02393-f002:**
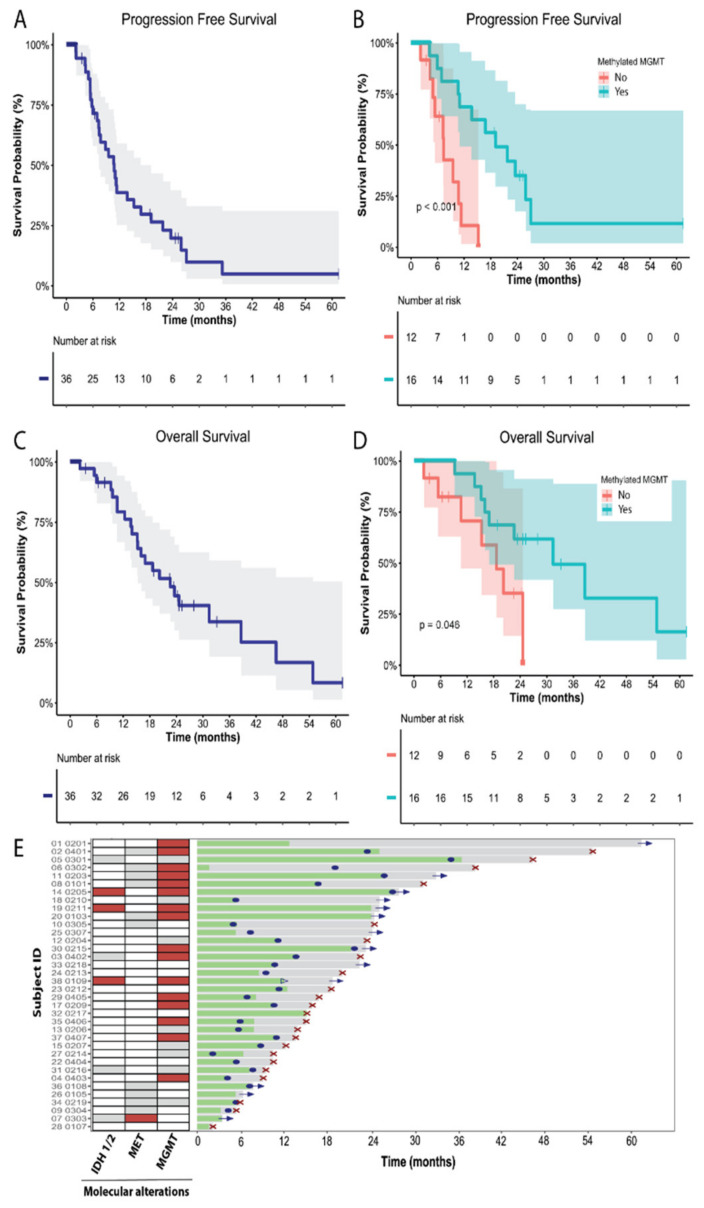
Crizotinib activity in the evaluable population for efficacy (*n* = 36 patients). (**A**) Progression-free survival (PFS) estimated by Kaplan–Meier. (**B**) PFS estimated by Kaplan–Meier stratified by the MGMT promoter methylation status. (**C**) Overall survival (OS) estimated by Kaplan–Meier. (**D**) OS estimated by Kaplan–Meier stratified by the MGMT promoter methylation status. (**E**) Swimmer plot showing the clinical evolution of each patient. Treatment period (green), follow-up (gray), progression disease (blue dot), exitus (red cross), and patient censored (blue arrow). Patients with presence of molecular alterations (IDH1/2 mutations, MET alterations, or MGMT methylation) are highlighted in red squares in the left panel. White squares indicate presence of WT or native genes. Gray squares on the left panel represent those patients that were not analyzed or not evaluable.

**Figure 3 cancers-14-02393-f003:**
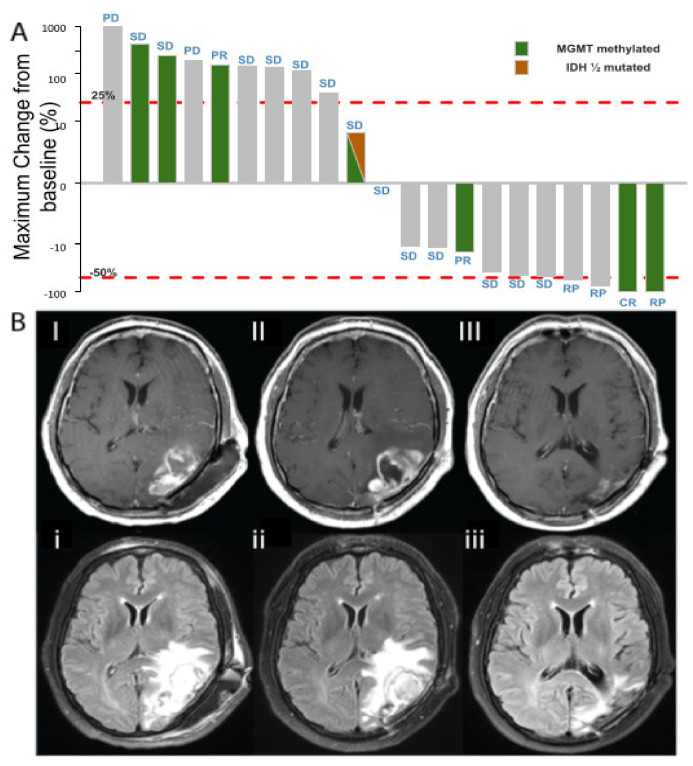
(**A**) Waterfall plot of MRI evaluation according to RANO criteria. Percentage change in tumor size at maximum reduction from baseline. Thirty-six patients were evaluable for tumor response. Red lines indicate the RANO cutoff for progressive disease (+25%) and PR (−50%). Note: one patient experienced a pseudo-progression on the first evaluation, while in the concomitant phase, having an increase of >100%. One month after the patient experienced a partial response (tumor shrinkage of 50% that lasted one month) and afterwards the patient remained stable. (**B**) Radiological evaluation of one patient in the dose escalation phase, at 250 mg/d dose level: 56-year-old male, partial resection in May 2016 (GBM, MGMT methylated). I. May 2016, post-operative MRI (I: Gd enhanced T1, i: FLAIR). II. August 2016, 4 weeks after RT and crizotinib, cycle 1 (II: Gd enhanced T1, ii: FLAIR). III. May 2017, maintenance therapy with crizotinib, cycle 9 (III: Gd enhanced T1, iii: FLAIR).

**Figure 4 cancers-14-02393-f004:**
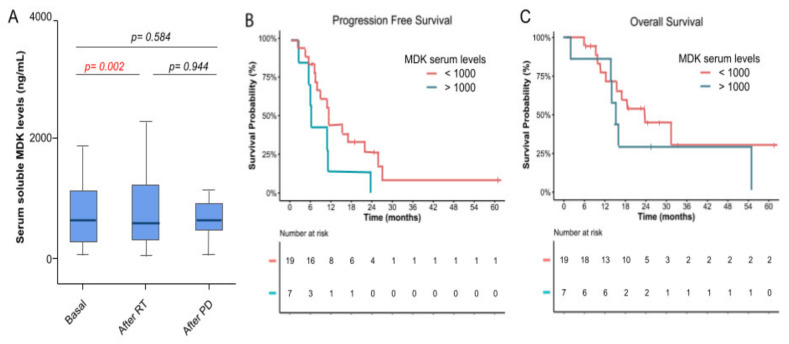
Correlation between MDK serum levels and efficacy. (**A**) Soluble MDK serum levels were quantified by ELISA at 3 timepoints throughout the study period: at baseline, after the concomitant phase of treatment, and after unequivocal progression of the disease. MDK levels are represented as a box plot with the median (dark line), the 25% and 75% quartiles (box), and 95% CI interval (cap lines). *p*-values were obtained by paired t-test. (**B**) PFS estimated by Kaplan–Meier stratified by the serum MDK levels. (**C**) OS estimated by Kaplan–Meier stratified by the serum MDK levels.

**Table 1 cancers-14-02393-t001:** Patient characteristics per protocol cohort (*n* = 36).

Patient Quantitative Characteristics		Median (Range)
Age (years)		51.4 (33.3–76.5)
KPS (%)		90 (70–100)
Barthel (%)		100 (75–100)
Mini-mental		29 (20–30)
Time from surgery to study treatment (weeks)		5.4 (4–8)
**Patient Qualitative Characteristics**		***n* (%)**
Gender	Male	16 (44.4)
	Female	20 (55.6)
Histological diagnosis	Glioblastoma	34 (94.4)
	Astrocytoma	1 (2.8)
	NE	1 (2.8)
MGMT methylation	Yes	16 (44.4)
	No	12(33.3)
	NE/UK	8 (22.2)
IDH1/2 mutations	Yes	3 (8.3)
	No	29 (80.6)
	NE/UK	4 (11.1)
EGFR amplification	Yes	12 (33.3)
	No	14 (38.9)
	NE/UK	10 (27.8)
ALK alterations	Yes	0 (0)
	No	26 (72.2)
	NE/UK	10 (27.8)
MET alterations	Yes	1 (2.8)
	No	25 (69.4)
	NE/UK	10 (27.8)
ROS alterations	Yes	0 (0)
	No	26 (72.2)
	NE/UK	10 (27.8)

**Table 2 cancers-14-02393-t002:** Grade ≥3 treatment-related adverse events classified by study phase, crizotinib dose, and grade.

Phase	Dose Escalation Phase				
**Crizotinib dose**	200 mg/QD	250 mg/QD	200 mg/BID	250 mg/QD	Any
**Grade**	Grade ≥3	Grade ≥3	Grade ≥3	Grade ≥3	Grade ≥3
**Number of patients *n* (%)**	3 (100%)	6 (100%)	3 (100%)	25 (100%)	37 (100%)
Fatigue	1 (33.3)	0 (0.0)	0 (0.0)	0 (0.0)	1 (2.7)
Transaminitis	1 (33.3)	1 (16.7) *	1 (33.3) *	5 (20.0)	8 (21.6)
Neutrophil count decreased	2 (66.7)	2 (33.3)	1 (33.3) *	1 (4.0)	6 (16.2)
Platelet count decreased	1 (33.3)	1 (16.7)	0 (0.0)	2 (8.0)	4 (10.8)
Constipation	0 (0.0)	0 (0.0)	1 (33.3) *	0 (0.0)	1 (2.7)
Lymphocyte count decreased	1 (33.3)	0 (0.0)	0 (0.0)	1 (4.0)	2 (5.4)
White blood cell decreased	0 (0.0)	0 (0.0)	0 (0.0)	1 (4.0)	1 (2.7)
Alanine aminotransferase increased	0 (0.0)	0 (0.0)	0 (0.0)	2 (8.0)	2 (5.4)

* Dose limiting toxicities (DLTs). Data cut-off at 10%. Toxicities reported here were causally related to any investigational medicinal product.

## Data Availability

Data from the GEINO-1402 clinical trial are publicly available in the Spanish registry of clinical trials, EudraCT registry (eudraCT number: 2014-000912-33) and at Clinicaltrials.gov (NCT number: NCT02270034).

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
