# Peer review of "Safety and Efficacy of Crizotinib in Combination with Temozolomide and Radiotherapy in Patients with Newly Diagnosed Glioblastoma: Phase Ib GEINO 1402 Trial"

_cancers, 2022, doi:10.3390/cancers14102393_

Round 1
Reviewer 1 Report
Thank you for an interesting study.
A multi-centre Phase 1b study utilising Crizotinib to inhibit ALK and c-MET is presented in newly diagnosed glioblastoma.
This is an interesting and well-designed study. There is a clear hypothesis and mechanism of action that is being targeted by the drug being investigated and is explained well.
Table 1 describes patient characteristics and it is noted that a proportion of patients have not had their full molecular profile (even MGMT, IDH1/2) evaluated. This is a minor shortcoming. Although numbers are small, it would be interesting to know if their was a different response in patients with different molecular markers (especially ALK and MET alterations), not least in view of an increasing trend towards molecular markers in diagnosing brain tumours.
Table 2 was a little hard to follow and the manuscript may benefit from it being presented better.
The manuscript will benefit from details of the ethical process being described (at present they appear briefly at the end).
Some minor errors of spelling and style (e.g. "astrocitoma", "minimental") may need attention.
However, this is a good study and my recommendation is for minor revision.
Author Response
Response to reviewers letter - GEINO1402
The authors want to acknowledge the reviewers for their kind and thoughtful revision.
Reviewer 1:
Thank you for an interesting study.
A multi-centre Phase 1b study utilising Crizotinib to inhibit ALK and c-MET is presented in newly diagnosed glioblastoma.
This is an interesting and well-designed study. There is a clear hypothesis and mechanism of action that is being targeted by the drug being investigated and is explained well.
Table 1 describes patient characteristics and it is noted that a proportion of patients have not had their full molecular profile (even MGMT, IDH1/2) evaluated. This is a minor shortcoming. Although numbers are small, it would be interesting to know if there was a different response in patients with different molecular markers (especially ALK and MET alterations), not least in view of an increasing trend towards molecular markers in diagnosing brain tumours.
Authors response:
Unfortunately, some patients did not have enough tumor sample to determine the molecular profile and could not been evaluated. As numbers are small, comparison regarding response is not feasible. Moreover, the study was not powered to observe differences in molecular profile. We have depicted the molecular profile of patients in the waterfall plot (Fig.3A) to address this in a descriptive manner.
Table 2 was a little hard to follow and the manuscript may benefit from it being presented better.
Authors response:
We are aware of the complexity of Table 2, as it contains detailed information on each dosing during the dose escalation phase and the expansion phase. We do agree that it will be good to simplify the information.
We have divided the any grade events and place them as supplementary online only material.
The manuscript will benefit from details of the ethical process being described (at present they appear briefly at the end).
Authors response:
Details on informed consent and ethical process have been included in the corresponding section.
Some minor errors of spelling and style (e.g. "astrocitoma", "minimental") may need attention.
Authors response:
The abovementioned spelling errors have been corrected.
However, this is a good study and my recommendation is for minor revision.
Authors response:
We deeply appreciate your comments and believe they all serve to improve the final quality of the manuscript. Thank you very much.
We deeply appreciate your comments and believe they all serve to improve the final quality of the manuscript. Thank you so much.
Sincerely yours,
The authors
April 28th, 2022
Reviewer 2 Report
The authors present the results of a phase 1b trial (dose escalation+expansion cohort) of RT/TMZ/crizotinib followed by TMZ/crizotinib. Safety and preliminary efficacy data is presented. Some patients experienced clear radiographic responses. The authors do not overstate their claims.
My critiques, detailed below, are all minor.
1) The Abstract states that patients received crizotinib monotherapy, however, the Methods state that patients received adjuvant crizotinib with temozolomide. This discrepancy should be rectified.
2) In Study Population, Gliadel should be capitalized.
3) In Table 1, astrocytoma is misspelled.
4) In Table 1, Yes should be added to each subgroup and 0% should be listed if the molecular abnormality was not detected in any patients.
5) In the Results, it would be worthwhile to present the results in the IDHwt MGMT unmethylated and IDHwt MGMT methylated populations, as the IDHwt population represents the contemporary classification of GBM (WHO 2021).
6) In the Discussion, it would be beneficial to add commentary on the potential superior CNS penetration of other ALK inhibitors as this could influence future directions in evaluating this approach in further clinical trials.
Author Response
Response to reviewers letter - GEINO1402
The authors want to acknowledge the reviewers for their kind and thoughtful revision.
Reviewer 2:
The authors present the results of a phase 1b trial (dose escalation+expansion cohort) of RT/TMZ/crizotinib followed by TMZ/crizotinib. Safety and preliminary efficacy data is presented. Some patients experienced clear radiographic responses. The authors do not overstate their claims.
My critiques, detailed below, are all minor.
1) The Abstract states that patients received crizotinib monotherapy, however, the Methods state that patients received adjuvant crizotinib with temozolomide. This discrepancy should be rectified.
Authors response:
The treatment schedule includes an initial concomitant phase in which patients received crizotinib in combination with radiotherapy and temozolomide; a second phase in which patients received crizotinib in combination with temozolomide; and a third phase in which patients received crizotinib only. The word monotherapy has been removed from the abstract for further clarity according to your suggestions.
2) In Study Population, Gliadel should be capitalized.
Authors response:
The word has been capitalized according to your suggestions Thank you
3) In Table 1, astrocytoma is misspelled.
Authors response:
The abovementioned spelling errors have been corrected.
4) In Table 1, Yes should be added to each subgroup and 0% should be listed if the molecular abnormality was not detected in any patients.
Authors response:
Table 1 has been modified according to the reviewer's suggestions.
5) In the Results, it would be worthwhile to present the results in the IDHwt MGMT unmethylated and IDHwt MGMT methylated populations, as the IDHwt population represents the contemporary classification of GBM (WHO 2021).
Authors response:
Only 3 patients had IDH mutated, so we consider that the impact of those over the overall population would be minor. The study did not exclude IDH mutations so the results should be reported for the full data set. We have reported the median OS for subgroups according to several molecular determinants in the results section, despite the numbers do not allow for statistically meaningful comparisons.
6) In the Discussion, it would be beneficial to add commentary on the potential superior CNS penetration of other ALK inhibitors as this could influence future directions in evaluating this approach in further clinical trials.
Authors response:
According to the reviewers suggestions the potential use of next generation ALK inhibitors has been discussed:
Moreover, next generation ALK inhibitors reach higher brain concentrations than crizotinib and may achieve even better control of the disease.
We deeply appreciate your comments and believe they all serve to improve the final quality of the manuscript. Thank you so much.
Sincerely yours,
The authors
April 28th, 2022